# Mesenchymal Stem Cell-Based Therapy for Rheumatoid Arthritis

**DOI:** 10.3390/ijms222111592

**Published:** 2021-10-27

**Authors:** Madina Sarsenova, Assel Issabekova, Saule Abisheva, Kristina Rutskaya-Moroshan, Vyacheslav Ogay, Arman Saparov

**Affiliations:** 1Department of Medicine, School of Medicine, Nazarbayev University, Nur-Sultan 010000, Kazakhstan; madina.sarsenova@nu.edu.kz; 2Laboratory of Stem Cells, National Center for Biotechnology, Nur-Sultan 010000, Kazakhstan; issabekova@biocenter.kz; 3Department of Family Medicine №1, Astana Medical University, Nur-Sultan 010000, Kazakhstan; saule_tabisheva@mail.ru (S.A.); rutskayakristina@gmail.com (K.R.-M.); 4Department of Arthrology, National Scientific Center of Traumatology and Orthopedics, Nur-Sultan 010000, Kazakhstan

**Keywords:** mesenchymal stem cells, rheumatoid arthritis, inflammation, cell therapy, cell preconditioning, immunomodulation

## Abstract

Mesenchymal stem cells (MSCs) have great potential to differentiate into various types of cells, including but not limited to, adipocytes, chondrocytes and osteoblasts. In addition to their progenitor characteristics, MSCs hold unique immunomodulatory properties that provide new opportunities in the treatment of autoimmune diseases, and can serve as a promising tool in stem cell-based therapy. Rheumatoid arthritis (RA) is a chronic systemic autoimmune disorder that deteriorates quality and function of the synovium membrane, resulting in chronic inflammation, pain and progressive cartilage and bone destruction. The mechanism of RA pathogenesis is associated with dysregulation of innate and adaptive immunity. Current conventional treatments by steroid drugs, antirheumatic drugs and biological agents are being applied in clinical practice. However, long-term use of these drugs causes side effects, and some RA patients may acquire resistance to these drugs. In this regard, recently investigated MSC-based therapy is considered as a promising approach in RA treatment. In this study, we review conventional and modern treatment approaches, such as MSC-based therapy through the understanding of the link between MSCs and the innate and adaptive immune systems. Moreover, we discuss recent achievements in preclinical and clinical studies as well as various strategies for the enhancement of MSC immunoregulatory properties.

## 1. Introduction

Rheumatoid arthritis (RA) is a chronic systemic disease that causes damage to joints, connective tissues, muscle, tendons, and fibrous tissue, and, as a result, has a major impact on society. Worldwide prevalence of RA is about 5 per 1000 adults. The disease is 2 to 3 times more frequently diagnosed in women than men, with a mean age of 55 years old [1,2]. The onset of the disease, also known as pre-RA phase, lasts months to years before clinical symptoms are presented and is subject to the presence of circulating autoantibodies, increased level of inflammatory cytokines and chemokines and altered cell metabolism [3]. The advanced form of the disease is characterized by severe and debilitating chronic pain that compromises patients’ quality of life. Inadequate management further results in disease progression, which ultimately leads to joint erosion, destruction and deformities. Previously, more than 50% of RA patients were disabled, incapable of serving on a full-time work basis, and were subject to increased mortality. However, a better understanding of disease pathophysiology and remarkable progress in the treatment of RA have led to the development of more efficient treatment approaches with the improvement of the disease activity control, the degree of pain and joint damage [1,4].

Regarding therapeutic approaches in RA, currently used drugs include glucocorticoids (GCs) and synthetic and biologic disease-modifying anti-rheumatic drugs (DMARDs) [5]. In addition to these, non-steroidal anti-inflammatory drugs (NSAIDs) are the most frequently used drugs for pain relief. GCs, in combination with NSAIDs or DMARDs, are used due to their potent anti-inflammatory effects. Among the above indicated conventional treatments, DMARDs demonstrated a high potential to reduce disease symptoms and prevent disease progression in patients with RA, however, they constitute high financial costs and exhibit serious side effects [6]. Additionally, despite significant pain reduction reported in numerous randomized controlled trials, many patients still experience clinically meaningful levels of remaining pain despite the treatment, and continue to be intolerant or resistant to these therapies [7,8].

Considering the limitations of conventional RA drugs, a modern cellular therapy based on mesenchymal stem cells (MSCs) may be regarded as an alternative strategy [9]. MSCs have attracted the attention of scientists and clinicians due to their capacity for self-renewal, tissue and organ regeneration and strong immunosuppressive properties. These characteristics enable suppression of the activity of pro-inflammatory cells of both innate and adaptive immune systems. It has been shown that MSCs are able to suppress the activation of natural killer (NK) cells and maturation of dendritic cells (DCs); inhibit the proliferation and function of T and B cells; promote macrophages’ polarization toward an anti-inflammatory phenotype; and induce the generation of T regulatory cells (Tregs) [10]. Moreover, it was demonstrated that the immunomodulatory effect of MSCs is mediated by both cell-cell contacts and through the secretion of soluble factors [11,12]. MSCs produce transforming growth factor-β (TGF-β), hepatocyte growth factor (HGF), prostaglandin E2 (PGE2), soluble form of protein HLA-G5, indolamine-2,3-dioxygenase (IDO), nitric oxide (NO) and interleukin-10 (IL-10) that are involved in the regulation and suppression of inflammatory responses [13]. All these mechanisms can contribute to controlling excessive inflammation in RA. To further improve the anti-inflammatory properties of MSCs for cell-based therapy, priming or preconditioning can be successfully applied. This approach allows for the use of bioactive substances (cytokines and growth factors), immune receptor agonists, hypoxia and 3D culturing [14]. Based on this wide range of immunomodulatory properties, the therapeutic potential of MSCs in RA treatment has been intensively studied in preclinical [15] and clinical studies [16,17,18,19,20,21,22]. Experimental animal models and human clinical trials demonstrated that MSCs have beneficial therapeutic effects in suppressing inflammation, bone erosion and joint destruction, as well as decreasing pannus formation through immunosuppression and immunomodulation.

In this review, we discuss current conventional therapies and the alternative therapeutic potential of MSCs for RA treatment by assessing mechanisms by which MSCs modulate the immune system and promote tissue repair. Various preconditioning strategies to enhance their therapeutic activity are reviewed as well.

## 2. Current Approaches in RA Treatment

The main goal in RA treatment is to achieve clinical remission or to reduce disease progression by inhibiting joint inflammation. Currently available conventional treatment methods include synthetic and biologic DMARDs and GCs [23].

DMARDs are the mainstay of RA therapy, which include heterogeneous drugs that inhibit disease progression and control symptoms [1]. Methotrexate (MTX), a conventional synthetic DMARD, is analogous to folic acid with anti-proliferative effects [24,25]. MTX causes the impairment of purine and pyrimidine metabolism, inhibits amino acid and polyamine synthesis and induces T cell and platelet apoptosis [26]. However, risks of skin cancer development and impairments in bone marrow as well as gastrointestinal, infectious, pulmonary and hematologic side effects have been observed in clinical practice [27,28].

Patients with moderate or severe disease should be initially treated with MTX as a monotherapy. In cases when disease inadequately responds to MTX, the treatment can be combined with a complementary drug, or fully replaced by other DMARDs if adverse effects are observed [29]. However, treatment with MTX is usually discontinued in less than 5% of patients due to side effects, which also can be reduced by prophylactic implementation of folates [30]. Alternative synthetic DMARDs include leflunomide, sulfasalazine, hydroxychloroquine and chloroquine. For patients with a mild disease course, hydroxychloroquine can be used as an initial therapy [31]. Leflunomide and sulfasalazine are also widely prescribed drugs for RA treatment, mostly in cases when patients have a contraindication to MTX [32]. Occasionally, a triple-drug therapy with MTX, sulfasalazine and hydroxychloroquine is applied [33]. Notably, MTX is preferred for use in patients because of its economical and therapeutic efficacy [34]. However, a combination of MTX with other drugs is reported to be a better treatment strategy than MTX alone [4,35].

The American College of Rheumatology (ACR) and European League Against Rheumatism recommend treatment with MTX in combination with short-term GC application, which is another potential anti-inflammatory drug for RA treatment for newly diagnosed patients [8,36]. The immunological effect of GCs is mediated by apoptosis of immature CD4+CD8+ thymocytes, and by the reprogramming of DCs to a tolerogenic state (tDCs). tDCs induce the generation of Tregs and increase macrophage phagocytosis of apoptotic cells [37,38,39]. However, side effects after GC application are more severe in comparison to other drugs. A dose increase in GCs causes ecchymosis, cushingoid features, parchment-like skin, leg edema, sleep disturbance and immunosuppression. Other adverse effects involve weight gain, epistaxis, glaucoma, depression, hypertension and diabetes [40,41]. Despite the adverse effects of GCs, the combination of MTX and GCs could reduce RA signs in about 25% of patients within 6 months of treatment. Moreover, in conjunction with systemic administration of GCs, intra-articular (IA) injections can prevent local joint inflammation [42].

About 30–50% of patients are unresponsive to conventional DMARDs. If a 2–6 month treatment with MTX mono- or combinational therapy is inadequate, biologic DMARDs should be added [36,43]. Biologic DMARDs include tumor necrosis factor (TNF) inhibitors, costimulation modifiers, IL-6 inhibitors and B cell depleting drugs. Commercially available biological drugs, such as etanercept (Enbrel^®^), infliximab (Remicade^®^), adalimumab (Humira^®^), golimumab (Simponi^®^) and certolizumabpegol (Cimzia^®^) are all TNF inhibitors that block cytokine signaling, reduce cell recruitment, normalize IL-6 expression level in serum and matrix metalloproteinase (MMP) expression levels in cartilage and bone, and as a result, slow bone destruction. Among the aforementioned biologic DMARDs, TNF inhibitors should be the initial drugs used in cases with an inadequate response to conventional synthetic DMARDs, and are often used in combination with other DMARDs, especially MTX [44,45]. Nonetheless, these drugs have serious side effects, such as increased risk of infections and neurological diseases, development of multiple sclerosis and lymphoma [46,47]. Additionally, TNF inhibition has previously resulted in the development of skin tumor Merkel cell carcinomas in patients affected by rheumatologic diseases [48]. Clinical trials with TNF inhibitors have revealed that a number of patients did not respond to treatment [49]. In this case, another DMARD, anakinra, that binds to IL-1 receptors and blocks inflammation, is considered for therapy. Anakinra is used in combination with other DMARDs or as a monotherapy, but the application is limited due to the risk of opportunistic and latent infections [50,51]. Anti-CD20 monoclonal antibody, rituximab, depletes B cells and is a typical medication for the treatment of lymphomas, leukemia and autoimmune disorders, and in RA patients, it can be added when there is an insufficient response to TNF inhibitors [52]. T cell activation can be blocked by abatacept, which is the fusion protein containing the domain of cytotoxic T lymphocyte-associated antigen 4 and prevents T cell activation by binding to CD80 and CD86 receptors on antigen-presenting cells (APCs), as well as blocks interaction between DCs and T cells. A clinical study of abatacept demonstrated significant results, however some patients were insensitive to this treatment, which was associated with the loss of CD28 expression on T cells [53,54]. Another biologic DMARD, tocilizumab, blocks IL-6 receptors and significantly reduces disease severity in RA patients who have not been effectively treated with traditional DMARDs [55]. A clinical study of anti-IL-17 antibody, secukinumab, and anti-IL-17RA antibody, brodalumab, has shown low response in RA patients in both cases [56,57]. Therapeutic efficacy of biologic DMARDs, when used as a monotherapy, is less effective compared to the combination with MTX [58].

It was shown that cytokines, such as TNF-α, IL-1, IL-6, IL-7, IL-15, IL-17, IL-18, IL-21, IL-23, IL-32, IL-33 and granulocyte-macrophage colony-stimulating factor (GM-CSF) were implicated in the pathogenesis of RA [59]. However, clinical trials with therapeutic strategies blocking IL-1, IL-18 or IL-17 have shown few benefits. On the other hand, TNF-α or IL-6 targeting therapy was successful in relieving symptoms and initiating disease remission [60]. Another approach in RA therapy is targeting small molecules. The Janus kinases (JAK) inhibitors are a type of targeted synthetic DMARD that recognize and regulate the activity of the JAK family of non-receptor tyrosine kinases, which transduce signals from several different cytokine receptors through the effects on the STAT family of transcription factors. Tofacitinib represents a targeted synthetic DMARD that inhibits IL-6 production by blocking JAK1 and JAK3 through the IL-6/gp130/STAT3 signaling pathway [61], and as a result, inhibits IL-17 and interferon-γ (IFN-γ) production and the proliferation of CD4+ T cells in patients with RA [62,63]. Another targeted synthetic DMARD, baricitinib, a JAK1/JAK2 inhibitor, was superior compared with the TNF-α antagonist adalimumab in patients with an inadequate response to the MTX [64]. Upadacitinib, which is a JAK1 inhibitor, significantly improves the efficacy of RA treatment in patients who are unresponsive to MTX or a TNF-α antagonist [65]. Thus, targeted synthetic DMARDs should be considered as a monotherapy or in combination with conventional synthetic DMARDs [66].

NSAIDs alongside GCs are commonly used as adjuvants to basic therapy. They are applied to decrease pain and inflammation during RA, however NSAIDs are not able to reduce bone and cartilage destruction [67]. NSAIDs are typically divided into two groups based on their chemical structure and selectivity: a group of non-selective NSAIDs, which inhibit both cyclooxygenase-1 (COX-1) and COX-2 and another group of COX-2 selective inhibitors. COX-1 plays a role in maintaining gastrointestinal mucosa lining, kidney function and platelet aggregation, whereas COX-2 is expressed during an inflammatory response. The most common NSAIDs applied in RA include acetylsalicylate, naproxen, ibuprofen and etodolac. Previously, NSAIDs were considered as first-line drugs, however low effectiveness in prevention of damage progression and side effects at high doses such as nausea, abdominal pain, ulcers and gastrointestinal bleeding, limited the implementation of these drugs [68].

Surgery is the final treatment approach for RA therapy in cases when the aforementioned nonsurgical methods are not sufficiently effective, which are becoming less frequent. Nowadays, various types of surgery are being applied, among them are tenosynovectomy, radiosynovectomy, arthroscopy, osteotomy and joint replacement. The final goal of surgical management is to relieve pain and restore joint function [69,70]. Table 1 summarizes the current approaches for RA treatment with the route of administration, mechanism of action and major side effects.

All of the described therapeutic strategies aim to maintain disease remission or low progression as well as reduce the risk of treatment, and thus, have relative efficacy. About 20–30% of moderate-to-severe RA patients are unresponsive to current treatment strategies [71]. In this regard, stem cell-based therapy with its immunomodulatory and immunosuppressive properties represents a promising approach in RA treatment.

## 3. Mesenchymal Stem Cells in RA Treatment

MSCs are adult multipotent stem cells with fibroblast-like morphology. Due to their immunomodulatory properties, MSCs are being considered for the treatment of autoimmune diseases [87]. The original definition of human MSCs is based on the presence of fibroblast-like morphology, with their ability to adhere to culture plastic and to differentiate into tissues of mesodermal origin, such as osteoblasts, chondrocytes and adipocytes. Furthermore, MSCs express CD73, CD90 and CD105 cell surface markers, but not hematopoietic and endothelial markers (CD14, CD34, CD45 and HLA-DR) [88,89,90,91]. The source of MSCs is a variety of tissues, including but not limited to, bone marrow (BM), gingiva, synovium, periosteum, adipose tissue (AT), dental pulp, umbilical cord (UC) and umbilical cord blood (UCB) [92,93,94,95,96,97,98]. In addition to their ability to differentiate into multiple cell lines, MSCs are capable of modulating innate and adaptive immune responses by alleviating the proinflammatory phenotype, particularly, through decreasing populations of DCs, macrophages, NK cells, B and T cells, and by promoting anti-inflammatory phenotype [11,99,100,101,102]. Depending on the environment, MSCs have the ability of polarizing and acquiring either pro-inflammatory (MSC1) or anti-inflammatory phenotypes (MSC2). In the presence of the inflammatory milieu (high levels of TNF-α and IFN-γ), which is generated by the immune cells, MSCs become activated and adopt an anti-inflammatory phenotype [11]. The mechanism of immune cell suppression by MSCs is mediated by secretion of a number of soluble factors, such as enzymes, cytokines and growth factors, including IDO, PGE2, NO, TGF-β1, HGF, hemoxygenase (HO), COX-2, IL-6 and IL-10. The IDO secretion is presumably induced by inflammatory cytokine IFN-γ [103]. The mechanism of action of IDO is mediated by conversion of an essential amino acid tryptophan to kynurenine, which impairs the synthesis of various cellular proteins and leads to the suppression of T cell proliferation. IDO is also considered to be involved in the generation of Tregs and tDCs induced by MSCs [104]. Moreover, factors produced by MSCs include nitric oxide synthase (iNOS), which induces the production of NO from macrophages thus inhibiting the proliferation, secretory and cytolytic functions of T cells. Both soluble factors function in the process of immunosuppression. However, it is demonstrated that iNOS mediates immunosuppression by mouse MSCs, while IDO plays a similar role in human MSCs [105].

Together with constitutive secretion of TGF-β by MSCs, the environment favors generation of Tregs [13,106,107,108,109,110,111]. In the absence of an inflammatory environment (low levels of TNF-α and IFN-γ), MSCs may adopt a pro-inflammatory phenotype and enhance T cell responses by secreting chemokines (e.g, MIP-1a and MIP-1b, RANTES, CXCL9 and CXCL10) that recruit lymphocytes to the sites of inflammation; these chemokines bind to CCR5 and CXCR3 expressed on T cells. The levels of immune suppressive mediators, such as IDO and NO, are low when the pro-inflammatory phenotype is adopted [12,112].

The role of apoptotic MSCs for therapeutic applications have been recently investigated. A study by Galleu and colleagues demonstrated that infused MSCs undergo extensive caspase activation and apoptosis in the presence of cytotoxic cells, which is a requirement for their immunosuppressive function, in both preclinical and clinical studies [113]. The mechanism is explained by the engulfment of apoptotic MSCs by phagocytes, and the IDO production, which is ultimately necessary for mediating immunosuppression. Similar results were reported by another group. The MCSs effect is based on the hypoxia-induced activation of caspase 3-mediated apoptosis, recruitment of immune cells at the transplantation site and their further engulfment by locally circulating macrophages [114].

Thus, the mechanism of immunomodulation by MSCs is regulated by both cell-cell interactions and paracrine effect via the secretion of soluble factors. Considering the abovementioned immunomodulatory properties, MSCs are being widely investigated as a promising tool for the treatment of autoimmune diseases, including RA.

### 3.1. In Vitro Studies

In vitro studies demonstrate that MSCs are capable of modulating functions of the innate immune system cells. These cells not only induce the inflammatory process but also activate the adaptive immune system, including T and B cells [115]. In RA, B cells differentiate to produce RF and ACPAs, and participate as APCs for T cell activation [116]. However, co-culturing AT MSCs with T cells, B cells and Tregs showed a two-fold increase in the number of CD4+CD25+FoxP3 Tregs. Additionally, MSCs inhibited CD3+T cell-mediated TNF-α secretion, upregulated IL-10 production and suppressed production of ACPAs by B cells [117].

The role of T cells is well described in an in vitro study by Vasilev and colleagues that confirmed the immunosuppressive capacity of the secretory factors produced by AT MSCs, to skew Th17/Treg balance towards Treg accumulation and also to downregulate major effector cytokine production involved in disease progression [118]. Through cell-cell interactions, human AT MSCs downregulated the production of TNF-α, IL-1β and IL-6 in mouse macrophages stimulated with lipopolysaccharide (LPS), and inhibited the proliferation of human primary T cells in response to mitogens [119].

Alternatively, MSCs have the ability to inhibit proliferation of effector memory T cells, which were found at high frequencies in the peripheral blood and synovial fluid of RA patients. These effector memory T cells are able to secrete proinflammatory cytokines, such as IFN-γ, IL-4 and IL-17. In contrast, MSCs are capable of modulating the immune response in RA by inhibiting both the proliferation of γδ effector T cells and their inflammatory cytokine production [120]. The mechanism of modulation is controlled by PGE2 driven by the existence of COX-2, which is released by MSCs [121]. Such immunosuppressive activity of MSCs can be explained by an appropriate inflammatory environment developed by the immune system cells which secrete pro-inflammatory cytokines.

Further investigations on the therapeutic potential of MSCs in the treatment of RA have been intensively studied recently in experimental animal models, mainly using a collagen-induced arthritis (CIA) model in mice, and will be discussed in the next section.

### 3.2. Preclinical Studies

Preclinical studies demonstrated that allogeneic MSC administration was more beneficial than administration of autologous MSCs [122,123]. MSCs are able to suppress inflammation both through interactions with the immune system cells and through the paracrine mechanisms. The schematic illustration represents the immunomodulatory effects of MSCs and their secreted factors in RA (Figure 1).

Cells of the innate immune system, such as macrophages, play a very important role in RA pathogenesis. It was shown that phagocytically active, proinflammatory HLA-DR+ macrophages can be detected in the synovium of RA patients [124]. These findings indicate that synovial macrophages of RA patients can influence T cell activation and subsequent migration and activation of B cells, thus generating an inflammatory response [125,126]. In this regard, MSCs have an impact on the macrophage polarization that maintains the balance between pro-inflammatory and anti-inflammatory phenotypes. MSCs constitutively produce IL-6, which either alone or in combination with LPS and/or pro-inflammatory cytokines, such as IFN-γ, polarizes pro-inflammatory M1 macrophages towards anti-inflammatory IL-10-producing M2 macrophages. M2 macrophages secrete high levels of IL-10 and TGF-β1 that suppress inflammation and promote tissue regeneration [11]. The polarization is likely initiated by a combination of cell-cell contact mechanisms and the production of soluble factors, such as IDO, PGE2, IL-10 and COX-2. For instance, UCB MSCs suppressed M1 macrophage proliferation and activated M2 macrophage production via TNF-α-mediated activation of COX-2 and TNF-stimulated gene-6 (TSG-6). Additionally, UCB MSCs downregulated nucleotide-binding domain, leucine-rich repeat pyrin 3 (NLRP3) inflammasome-mediated IL-1β secretion and caspase-1 production in macrophages through IL-1β feedback loop in CIA mice [127]. Furthermore, osteoclast activity is upregulated in RA that further induces severe bone destruction, while in healthy conditions, osteoblast and osteoclast activity is balanced, promoting a normal level of bone formation and resorption [128]. Moreover, TNF-α and IL-1β that are secreted by pro-inflammatory macrophages cause the activation of synovial fibroblasts, which secrete the receptor activator of NF-κB ligand (RANKL) and macrophage colony-stimulating factor 1 (M-CSF). These factors are indispensable in osteoclast formation [129]. On this matter, AT MSCs inhibited both RANKL-induced osteoclastogenesis and decreased the osteoclast precursors in bone marrow, leading to the prevention of systemic bone loss in CIA mice [130].

DCs are also cells of the innate immune system that serve as a professional APC. They are characterized by a stellate morphology and high expression of MHC class II, as well as by their capacity to uptake antigens and migrate to draining lymph nodes to prime naive T cells [131]. However, in RA, DCs are also responsible for inducing inflammation by presenting antigens to autoreactive T cells with subsequent production of cytokines, which stimulate T-helper differentiation. The pannus developing in RA likely occurs with the involvement of circulating DCs via chemokine signaling. Although direct evidence is still lacking, several chemokine receptors expressed on DCs have been implicated in the pathogenesis of CIA. These include CX3CR1, CCR9, CXCR4 and CCR2 [131]. Additionally, in a recent study, the inhibition of CCR7-mediated DCs migration towards draining lymph nodes by administration of FTY720, an immune-modulator that is a chemical derivative of myriocin, was also shown to ameliorate CIA [132]. More detailed information on the mechanism of action of MSCs on DCs is mainly known from other autoimmune diseases, and is based on their ability to inhibit maturation of DCs and enhance the generation of tDCs by reducing Toll-like receptor (TLR) activation and suppressing IL-12 production by DCs [133]. Moreover, MSCs and tDCs have demonstrated a synergistic immunosuppressive effect in a CIA model by polarizing Th cells, inhibiting proinflammatory cytokine production and reducing cartilage degeneration [134].

Similar results on the effects of MSCs on B cells confirmed the data obtained from in vitro studies. MSCs significantly contributed to the inhibition of plasmablast generation and influenced B cell differentiation, which was mediated by IL-1 receptor antagonist (IL1RA) secreted by MSCs [99]. Based on published data, the mechanism of action of MSCs on B cells is mediated not only by direct cell-to-cell interactions, but also by soluble factors produced by MSCs.

T cells have a greater impact and play a crucial role in RA pathogenesis. MSCs have the ability to control the proliferation, differentiation and activity of T cells and reduce the production of pro-inflammatory cytokines. Recent studies also confirmed the regulatory capacities of MSCs. For instance, some studies demonstrated that the application of MSCs significantly decreased pro-inflammatory cytokine levels (IL-1β, IL-6), whereas the expression level of anti-inflammatory cytokine (IL-10) increased. Furthermore, a number of Tregs also significantly increased after the human UCB MSCs treatment [135,136]. The analysis of CD4 T cell populations from CIA mice after treatment with human embryonic stem cell-derived (hESC) MSCs ameliorated CIA by inducing IFN-γ+ Th1 cells and IDO1 and showed an increase in the number of FoxP3+ Tregs [137]. Furthermore, human hAT MSCs treatment induced the expansion of Tregs in the peripheral blood and in the spleen [119]. Additionally, administration of hUC MSCs in a mouse CIA model prevented disease development by reducing the frequency and functions of T follicular helper cells (Tfh) through IDO activity [138]. Thus, MSCs can also suppress the differentiation of Tfh toward effector subsets, such as Tfh1, Tfh2 and Tfh17, and as a consequence, can reduce the production of autoreactive antibodies.

The pathogenic role of cytotoxic CD8+ T cells (CTLs) in RA has not been broadly investigated. However, there is evidence of a direct correlation between RA severity and the number of CTLs in the joint, where a high frequency of CD8+ T cells at the inflammation site has been observed [139]. Additionally, a recent study by Vohra and colleagues confirmed that UC MSCs downregulated the functions of activated CD4+ and CD8+ T cells from both the peripheral blood and synovial fluid of RA patients, suppressed the secretion of pro-inflammatory cytokines and induced the expansion of Tregs. Intraperitoneal (IP) injection of UC MSCs in CIA rats clearly indicated a sustained impact in terms of slowing down the progression of disease [140]. Studies showed that IV administration of hAT MSCs in CIA mice decreased GM-CSF expressing CD4+ T cells in blood and spleen, which are considered to be key effector cells in RA pathophysiology [141]. In addition, the above evidence leads to the conclusion that RA is a Th1/Th17-induced disease, and Tregs have the capacity to modulate inflammatory processes. MSCs in this regard have a balancing effect between Th cells and Tregs that is modulated by soluble factors, such as IDO, PGE2, IL-10, NO and HGF [142,143].

Thus, it can be generalized that a sufficiently large array of data on the application and the possible immunomodulatory effects of MSCs in RA are under investigation. The existing research data differ from each other due to various specifying variables. These include the origin of MSCs (human or mouse), tissue source, route of administration, timing of treatment, number of repetitions, dosage and mouse/rat strains, which are all critical and have different effects on the therapeutic outcome. The in vivo studies related to the effects of MSCs on the cells of innate and adaptive immunity described in this section are summarized in Table 2.

It was recently demonstrated that syndecan-3 (SDC3), which is the largest cell surface molecule from the syndecan family, plays an important role in inflammatory disorders such as RA. MSCs derived from SDC3 knockout mice possessed enhanced adhesion to collagen type I and AKT pathway hyperactivation. This evidence suggests that SDC3 targeting might be a promising therapeutic strategy for RA treatment [145]. In conclusion, meta-analysis data demonstrated that MSCs were effective in RA treatment in animal models. It suggests that in preclinical studies, MSCs have consistently exhibited therapeutic benefits. Authors also reported that human UC MSCs led to significant improvements in clinical and histological scores. These data suggest that human UC MSCs could serve as the most appropriate cell source for RA treatment application [15].

### 3.3. Clinical Studies

Several MSC-based therapies for the treatment of RA were studied in 18 clinical trials [146]. Currently, nine of these trials are still in progress, and the remaining clinical studies were completed and published. Clinical trials investigating the therapeutic potential of MSCs from various tissue sources were mainly focused on the evaluation of the safety and efficacy of the transplantation of MSCs in RA (Table 3). In this section, we describe some of the major clinical studies initiated during the last 10 years.

The first clinical study was started in 2010 by Ra and colleagues [17]. They investigated the safety and efficacy of IV and IA infusion of autologous AT MSCs in RA patients. The patients were split into three-dose regimens with different amounts of AT MSCs. The first group received two separate IV doses of 3 × 10^8^ AT MSCs. The second group received two injections of AT MSCs: (1) intravenous (IV) injection of 2 × 10^8^ and IA injection of 1 × 10^8^ AT MSCs into finger, wrist, elbow and knee joints; (2) IV injection of 3.5 × 10^8^ and IA injection of 1.5 × 10^8^ AT MSCs. The third group received four IV injections of 2 × 10^8^ AT MSCs in intervals of one month. The results of this study demonstrated that autologous AT MSCs are safe and provide clinical improvement in RA patients.

In another large randomized multicenter clinical trial, 53 patients with refractory RA were recruited for evaluation of efficacy of different dosages of allogeneic AT MSCs [16]. RA patients were divided into three groups and received IV injection of allogeneic AT MSCs with doses of 1, 2 or 4 × 10^6^ cells/kg of body weight, three times with an interval of 1 week. The results of this study demonstrated that IV injection of allogeneic AT MSCs resulted in 2% clinical improvement in 20–45% of RA patients, according to the criteria of the ACR after 1 month regardless of the administered dose of MSCs. This therapeutic effect persisted after 3 months in 15–25% of RA patients receiving MSCs, but not in the placebo control group. It was concluded that the use of MSCs was well tolerated without manifestation of dose-dependent toxicity.

From 2011 to 2013, 30 RA patients were recruited for a randomized, triple-blind placebo-controlled phase 1/2 clinical trial to study the safety and tolerability of IA injection of autologous BM MSCs in RA patients [18]. The results published in 2018 showed that MSCs administration does not exert any adverse effects in RA patients. Moreover, it was revealed that in comparison to the patients in the placebo group, patients who received IA injection of BM MSCs demonstrated superior clinical results according to the Western Ontario and McMaster Universities Arthritis Index (WOMAC), visual analogue scale (VAS), time to jelling and pain-free walking distance up to 12 months. Based on these data, the authors have suggested that IA knee injection of BM MSCs is generally safe and well tolerated in RA patients.

Recently, Ghoryani and colleagues completed a successful clinical trial on the effects of IV administration of autologous BM MSCs on the various immunological, clinical and para-clinical indicators that are associated with the pathogenesis of RA in patients with refractory RA [19]. They showed that a single IV injection of 1 × 10^6^ BM MSCs/kg resulted in a significant decrease in Th17 cell number, disease activity score 28-erythrocyte sedimentation rate (DAS28-ESR) and VAS at 12 months after MSC therapy. However, no significant changes were observed in serum C-reactive protein (CRP) and anti-cyclic citrullinated antibody (anti-CCP) levels in refractory RA patients after injection of autologous BM MSCs. Taken together, these clinical data suggested that autologous BM MSCs can significantly ameliorate the severity and activity of refractory RA.

In 2013, a group of researchers performed a phase 1/2 clinical trial to evaluate the safety and efficacy of IV injection of allogeneic hUC MSCs in patients with active RA [20]. In the study, 172 RA patients who failed to respond to conventional treatment were enrolled. The control group of patients received culture medium without UC MSCs. The experimental group of patients received a single dose of 4 × 10^7^ UC MSCs. All groups of patients received DMARD treatment. The results of the clinical study showed that UC MSCs treatment did not induce any adverse effects and resulted in the following clinical improvements: a moderate reduction in inflammatory cytokines and chemokines, an increase in percentage of Tregs in peripheral blood and upregulation of IL-4-producing Th2 cells. In addition, a significant disease remission was observed by the ACR improvement criteria, the DAS28 score and the Health Assessment Questionnaire (HAQ), which were maintained for 3–6 months without repeated IV injection of UC MSCs. Moreover, an additional clinical study demonstrated that UC MSCs treatment can exert long-term beneficial effects in RA patients for up to 3 years [21]. Thus, this clinical trial showed that IV administration of allogeneic UC MSCs in combination with DMARDs was safe and effective in ameliorating disease activity in refractory RA patients, compared to the control group that received culture medium without UC MSCs.

In comparison to the aforementioned clinical studies, Korean scientists from KangStem Biotech performed a single IV infusion of 2.5 × 10^7^, 5 × 10^7^ or 1 × 10^8^ of allogeneic UCB MSCs to RA patients that did not previously receive any biologic drugs [22]. A phase Ia clinical trial showed that no patients exhibited any serious adverse events and abnormalities in hematologic profiles during and after the treatment. It was revealed that IV infusion of UCB MSCs (1 × 10^8^ cells per patient) significantly reduced the levels of inflammatory cytokines in peripheral blood of RA patients at 24 h, and the mean DAS28-ESR, HAQ and VAS score declined at week 4. Despite some limitations, such as a relatively small number of recruited patients and the short duration of the follow-up period after UCB MSCs treatment, obtained clinical evidence demonstrated that a single high dose of allogeneic UCB MSCs was absolutely safe and effective for the treatment of refractory RA patients.

Thus, the early clinical studies described in this section indicate that both autologous and allogeneic MSC transplantation is safe and effective for treatment of refractory RA patients. No serious adverse effects have been reported in any of the RA patients during these clinical trials. The patients who received MSC treatment showed a moderate reduction in serum inflammatory markers, symptomatic improvement and significant disease remission. Moreover, it has been reported that the therapeutic effects after MSC treatment of RA patients can be maintained for up to 3 years with a stable clinical outcome, indicating the long-term safety and efficacy of MSC-based therapy.

Despite the promising results of clinical trials, there are some limitations in RA treatment with MSCs. Firstly, most studies have been conducted on RA patients enrolled from a single center, and sometimes without inclusion of a placebo control. In addition, patient enrollment in some clinical trials for evaluation of safety and efficacy was low. In some cases, MSC-treated groups included one or three patients. Therefore, to confirm the current clinical data on the efficacy of MSC therapy, a multiple-center, controlled trial should be conducted with the enrollment of a large number of RA patients. Secondly, currently there is not yet an optimal protocol for RA treatment with MSCs. This is due not only to the discrepancy of MSC sources, but also to the different routes of administration, treatment regimens and dosing used in the clinical studies. Taken together, these discrepancies in study designs have introduced difficulty when comparing therapeutic outcomes. Nevertheless, most clinical studies have revealed that regardless of the route of administration, the therapeutic efficacy of MSCs is achieved at a dosage of at least 1 × 10^6^ cells/kg of body weight after single or multiple injections. Although a dose-dependent relationship between MSC treatment and response has shown therapeutic effects, there is yet to be a well-defined and effective therapeutic window for RA with MSCs. Thus, treatment regimens and dosage adjustments must be studied thoroughly in future clinical studies. Thirdly, it is known that MSC therapy is an expensive treatment procedure compared to DMARDs or biologics. However, during prolonged drug therapy, 15–40% of RA patients develop resistance to these drugs and experience an increase in the incidence of side effects that adversely affect the patient’s health. Completed clinical studies showed that infusion of MSCs is a safe and effective approach for treating RA patients and does not cause serious adverse effects. In this regard, MSCs may constitute a therapeutic vehicle for RA patients who are resistant to DMARDs. In addition, enhancement of immunomodulatory and anti-inflammatory properties of MSCs using the strategies of cell preconditioning may not improve therapeutic efficacy, but it may reduce the cost of manufacturing MSCs for an effective treatment for RA patients. Thus, future studies using the innovative cell technology can be the key to increasing scientific data on the therapeutic efficacy of MSC-based therapy.

### 3.4. Strategies to Improve the Therapeutic Effects of MSCs

As the therapeutic effects of MSCs have already been confirmed, researchers and clinicians are interested in the development of new strategies to increase the potential of MSCs in clinical applications for RA treatment [147]. To date, several strategies have been proposed to enhance the immunomodulatory and anti-inflammatory properties of MSCs in RA [14]. Among them are coculture methods, growth factors and cytokines, receptor agonists, hypoxia, autophagy and modifications in culture methods such as 3D culturing. An entirely different approach is the genetic modification of MSCs (Figure 2) [148,149,150,151]. Genes, which are involved in the increase in cell survival rate, immunomodulation and regeneration, are modulated by genetically engineered constructs, such as viral vectors or plasmids [152].

In the strategy suggested by Lim and colleagues, the combined application of MSCs and IL-10-producing Tregs was more effective in suppressing inflammatory responses in joints and preventing the development of destructive arthritis in mice compared to the uncombined cell therapy [153]. Another promising strategy for the improvement of the therapeutic potential of MSCs is culturing the cells as 3D spheroids. Although MSC preconditioning in the form of 3D culturing has not yet been tested in an RA model, obtained data from several studies support the application of this strategy. 3D spheroid culture, in contrast to adherent monolayer culture, mimics a physiologically relevant microenvironment through intensive cell-cell and cell-matrix interactions. A number of studies have shown that culturing MSCs in a 3D microenvironment significantly increased their immunomodulatory and anti-inflammatory properties, apparently through upregulated TSG-6 and COX-2 expressed by MSC spheroids [154,155]. Furthermore, it was shown that both MSC spheroids and MSCs from spheroid more effectively suppress TNF-α production by LPS-stimulated peritoneal macrophages in vitro and inflammatory reactions in an in vivo mouse model of zymosan-induced peritonitis. In addition, it has been demonstrated that the 3D spheroid culture of MSCs produce higher levels of PGE2, TGF-β1, IL-6 and IDO in contrast to a traditional 2D monolayer culture, confirming the activation of the immunomodulatory capacities of MSCs in a 3D environment [156].

Another strategy to improve the therapeutic effects of MSCs is to target the immune receptor agonists. TLRs expressed in MSCs have the ability to recognize potential threatening signals, and for this reason TLR3 and TLR4 have been relevant targets in implementing this strategy [157]. TLR3 and TLR4 proteins were targeted to improve cellular MSC properties by ligation of their agonists, polyinosinic:polycytidylic acid (poly I:C) and LPS, respectively. Once TLR3 is ligated, it generates further activation of downstream cascades. Poly(I:C) stimulates the Notch signaling pathway and enhances immunomodulatory properties, such as Treg promotion and impairment of Th1/Th17 cell expansion. In addition, TLR3 activation is shown to be involved with PGE2 expression that causes upregulation of an immunosuppression factor in BM MSCs [158,159]. Interesting data were reported by Moases and colleagues who demonstrated that caffeine primed MSCs can reduce the production level of pro-inflammatory cytokines, such as IFN-γ, IL-6 and IL-1β, in vitro and significantly downregulate the disease status in vivo compared to treatment with MSCs alone. Caffeine preconditioned MSCs caused a decrease in CRP, NO, myeloperoxidase and TNF-α levels in serum and conversely led to a significant increase in the levels of IL-10 than in the control group [160].

Additional prospective approaches to increase the immunomodulatory effects of MSCs for RA treatment are hypoxia and autophagy. Recent results confirm the prospective application of hypoxia and autophagy conditions for future MSC-based therapy for RA treatment. Some studies have shown that autophagy has an important role in protecting from ROS generated in MSCs after oxidative stress or irradiation [161]. For activation of autophagy in MSCs, pretreatment with starvation and mTOR inhibitor rapamycin are usually used. Hypoxic preconditioning increases the immunomodulatory effects of MSCs due to the upregulation of secreted immunoregulatory factors, including PGE2 and IDO [162,163,164,165,166]. Similarly, it has been shown that priming human MSCs with hypoxia or IFN-γ resulted in immunosuppressive effects on CD4+ and CD8+ T cell proliferation. However, combined application of IFN-γ and hypoxia synergistically inhibited T cell proliferation and significantly increased IDO and HLA-G expression. Treatment by one of the mentioned priming factors alone showed less of an effect, suggesting that the immunomodulatory effects of MSCs could be enhanced by the combinatory application of pro-inflammatory cytokines and hypoxia together [167].

Another promising strategy of MSC preconditioning is priming by pro-inflammatory cytokines. This approach is based on the fact that MSCs act as sensors of inflammation, and as a consequence, can significantly enhance their immunomodulatory and immunosuppressive properties [11]. In the presence of high levels of proinflammatory cytokines, MSCs are activated and acquire a pronounced immunosuppressive phenotype by producing high levels of anti-inflammatory mediators, such as PGE2, IDO, TGF-β, HGF, NO and heme oxygenase. Given this phenomenon, cell preconditioning with high concentrations of proinflammatory cytokines is used to enhance the immunosuppressive properties of MSCs. For example, some studies have demonstrated that when compared with untreated MSCs, MSCs preconditioned with IFN-γ and/or IL-1β more effectively suppressed T cell proliferation, CD8+ T cell degranulation, NK cell and macrophage activation and the production of proinflammatory cytokines (TNF-α, IFN-γ and IL-2) by activated T cells [168,169,170,171]. On the other hand, treatment of MSCs with IFN-γ led to an increase in the number of Tregs and the secretion of anti-inflammatory and immunoregulatory cytokines [172]. Sivanathan and colleagues showed that the preconditioning of human MSCs with IL-17A is as effective as treatment with IFN-γ for suppressing the activation and proliferation of T cells and production of Th1 cytokines (TNF-α, IFN-γ and IL-2). In addition, IL-17A-treated MSCs significantly contributed to the generation of induced Tregs [173]. Preconditioning of MSCs with TNF-α, IL-1α or IL-1β as separate agents also enhances the immunomodulatory effects of MSCs. Furthermore, a number of studies have shown that the combined preconditioning of MSCs using IFN-γ with one of the aforementioned proinflammatory cytokines can further enhance the immunosuppressive effects of MSCs [151].

## 4. Conclusions

MSCs have been extensively employed in treatment in experimental animal models for inflammatory and immune disorders. However, due to their robust abilities to exert immunomodulatory effects, they have been most therapeutically efficient in the treatment of autoimmune diseases, such as graft versus host disease, lupus erythematosus, multiple sclerosis and RA. The capacity of MSCs to reduce T cell proliferation and to suppress the inflammatory infiltrates and cytokines has been well documented. Additionally, it has been identified that modulatory mechanisms are mediated by multiple interactions, including cell-cell contacts and paracrine effects. Currently, MSC-based therapy is widely applied in clinical practice for the treatment of various diseases. According to the NIH, more than 350 clinical studies on MSC-based therapy are currently underway, and only about 10 of them are associated with RA treatment. In the context of RA, various immune cells, such as macrophages, DCs, NKs, B cells and T cells, with their numerous subtypes, are involved in inducing a complex immune response. Based on the complexity of disease pathogenesis, MSCs can be considered a promising alternative approach with the capacity to provide strong immunomodulatory properties for RA treatment. However, the environment, which is created by the cells of both innate and adaptive immunity and their secreted factors, influences the ability of MCSs to acquire either an anti- or pro-inflammatory phenotype, which can be reversed depending on the environment. Thus, prospective clinical application of MSCs in RA treatment needs to be further investigated. Moreover, it is important to note that the strategy of MSC priming, which is aimed to promote an anti-inflammatory phenotype, enhances the immunomodulatory potential for further therapeutic applications.

## Figures and Tables

**Figure 1 ijms-22-11592-f001:**
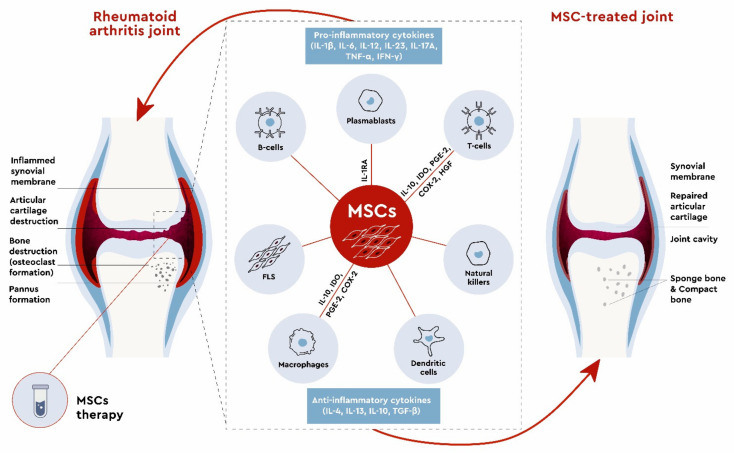
Immunomodulatory effects of mesenchymal stem cells and their secreted factors in rheumatoid arthritis.

**Figure 2 ijms-22-11592-f002:**
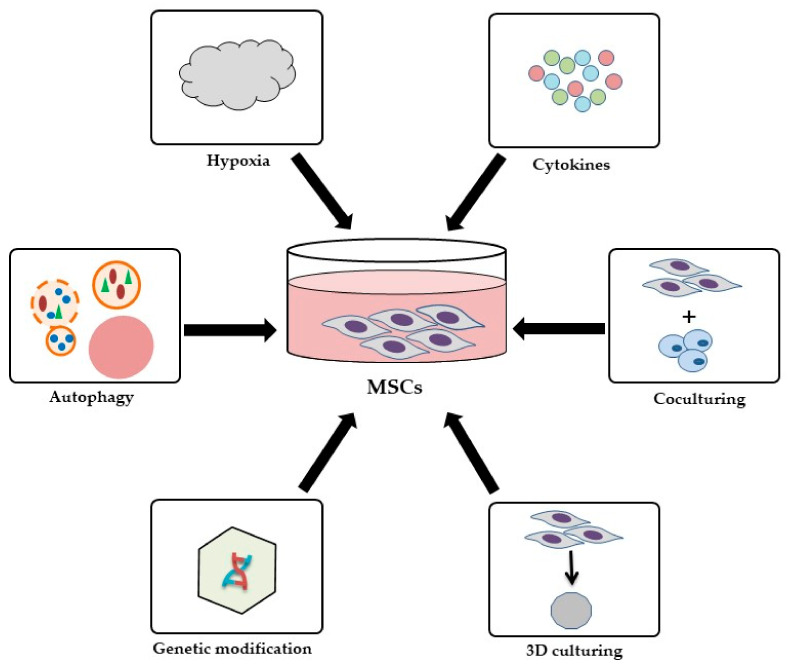
Different approaches to enhance immunomodulatory and anti-inflammatory properties of MSCs in RA.

**Table 1 ijms-22-11592-t001:** Summary of current approaches for RA treatment.

Drug	Example	Administration/Dose	Mechanism of Action	Side Effects	Reference
Conventional Synthetic DMARDs	MTX	Orally or intravenous (IV) injection (15 mg), single subcutaneous (SC) or intramuscular (IM) injection (15–25 mg/week)	Impairs purine and pyrimidine metabolism, inhibits amino acid and polyamine synthesis	Skin cancer and gastrointestinal, infectious, pulmonary and hematologic side effects, bone marrow impairments	[27,28]
	Leflunomide	Orally (50 mg/week or 10 mg/day)	Inhibits dihydroorotate dehydrogenase enzyme leading to inhibition de novo synthesis of pyrimidine nucleotides	Dyspepsia, nausea, abdominal pain and oral ulceration	[72]
	Sulfasalazine	Orally (500 mg/daily or 1 g/day in 2 divided doses up to a maximum of 3 g/day in divided doses)	Suppresses the transcription of nuclear factor-κB (NF-κB) responsive pro-inflammatory genes including TNF-α	Nausea, vomiting, anorexia, dyspepsia, male infertility (reversible), headache and skin rash	[73]
	Hydroxychloroquine	Orally (400 mg/day over a 30-day period)	Increases pH within intracellular vacuoles and alters processes such as protein degradation by acidic hydrolases in the lysosome, assembly of macromolecules in the endosomes and post-translation modification of proteins in the Golgi apparatus	Retinal toxicity, neuromyotoxicity	[74,75]
Biologic DMARDs	Etanercept, Infliximab, Adalimumab, Golimumab, Certolizuma-bpegol	Etanercept—SC injection (50 mg/week or 25 mg/twice a week); Infliximab—SC injection (3–10 mg/kg every 4–8 weeks); Adalimumab—SC injection (25 mg/twice a week); Golimumab—SC injection (50mg/month); Certolizumab pegol—SC injection (400 mg at weeks 0, 2 and 4, followed by 200 mg every 2 weeks)	Blocks the biological activity of TNF	Infections, neurological diseases, development of multiple sclerosis and lymphomas	[76]
	Anakinra	SC injection (75–150 mg or 0.04–2 mg/kg)	Binds to IL-1 receptors	Opportunistic and latent infections	[77]
	Rituximab	IV injection (1 gm twice separated by 2 weeks) with MTX and IV corticosteroid premedication	Anti-CD20 monoclonal antibody	Hypogammaglobulinemia, rarely serious infectious events	[78]
	Abatacept	IV injection (2–10 mg/kg on days 1, 15 and 30, and then every 4 weeks)	Contains the domain of cytotoxic T lymphocyte–associated antigen 4 (CTLA-4), blocks interaction between DCs and T cells	Serious infections, increased risk of certain malignancies	[79]
	Tocilizumab	IV injection (8 mg/kg once every 4 weeks) or SC injection (162 mg/week)	Blocks IL-6 receptor	Serious infections, major adverse cardiovascular events, cancers, diverticular perforations, hepatic diseases, rarely lethal	[80]
	Secukinumab	SC injections (25–300 mg)	Primarily targets IL-17A	Nasopharyngitis or infections of the upper respiratory tract, mild-to-moderate candidiasis	[81]
	Brodalumab	SC injection (70–210 mg)	Prevents the nuclear factor kappa light chain enhancer of activated B cells, IL-6, IL-8, COX-2, MMPs and GM-CSF	Nasopharyngitis, upper respiratory tract infections, arthralgia, back pain, gastroenteritis, influenza, oropharyngeal pain, sinusitis	[82]
Targeted Synthetic DMARDs	Tofacitinib	Orally (5 mg/twice daily)	Blocks Janus kinases (JAK1 and JAK3)	Cardiovascular events, neutropenia and lymphopenia, risk of infection (viral reactivation, herpes virus reactivation, opportunistic infections)	[83]
	Baricitinib	Orally (4 mg/day or lower dosage 2 mg/day)	Inhibits JAK1/JAK2	Hyperlipidemia, viral reactivation, deep venous thrombosis and pulmonary embolism event	[84]
	Upadacitinib	Orally (15 mg/day or 30 mg/day)	Inhibits JAK1	Upper respiratory tract infection, nasopharyngitis, and urinary tract infections, gastrointestinal perforation	[85]
GCs	Dexame-thasone, be-tamethasone, triamcinolone, prednisone, prednisolone	The addition of GCs, to either standard DMARD monotherapy or combinations of synthetic DMARDs with low-dose GCs (< 7.5 mg/day) or high-dose GCs (up to 15 mg/day)	Directly activates or represses gene transcription	Ecchymosis, cushingoid features, parchment-like skin, leg edema, sleep disturbance, immunosuppression, weight gain, epistaxis, glaucoma, depression, hypertension, diabetes	[40,41,86]

**Table 2 ijms-22-11592-t002:** Preclinical studies for RA treatment with MSCs.

RA Model	Source and Tissue Origin of MSCs	Route of Administration/Number of Repetitions	Dosage	Mechanism of Action	Therapeutic Outcome	Reference
CIA in DBA1/J mice	hUCB MSCs	IP injections for 5 days after the RA score reached 3 or more	1 × 10^6^ cells	hUCB MSCs polarized M1 macrophages toward M2 phenotype through TNF-α-mediated activation of COX-2 and TSG-6	Amelioration of the severity of CIA	[127]
CIA in DBA/1J mice	hBM MSCs	IP injection on day 22 after primary immunization	2 × 10^6^ cells	hBM MSCs inhibited RANKL-induced osteoclastogenesis	Amelioration of inflammation-induced systemic bone loss in CIA	[130]
CIA in DBA1/J mice	hUC MSCs	IV injection on day 28 after RA score reached 1 or more	1 × 10^6^ cells	hUC MSCs reduced number and downregulated function of Tfh cells in the spleen accompanied with decreased Th1 and Th17 cells	Prevention of CIA progression	[138]
CIA in DBA/1OlaHsd mice	hESC MSCs	Single-dose IP injection on the day of immunization (prophylaxis) or with three doses of hESC MSCs every other day starting on the day of arthritis onset (therapy)	1 × 10^6^ cells	hESC MSCs increased the number of FoxP3(+) Tregs and IFN-γ^+^ Th1 cells but not Th17, additionally induced the expression of IDO1 in inguinal lymph nodes	Reduction of disease progression and severity of CIA	[137]
CIA in DBA1/J mice	hUCB MSCs	IV injection of three different doses every 2 weeks, overall, three times	1 × 10^6^ cells, 3 × 10^6^ cells, 5 × 10^6^ cells	hUCB MSCs decreased IL-1β and IL-6 levels; concentration of 5×10^6^ hUCB MSCs increased the level of IL-10 production and the expansion of Tregs	Alleviation of RA symptoms in a CIA model	[135]
CIA in DBA1/J mice	hUC MSCs	IV injection after 24 days after RA induction	2 × 10^6^ cells	hUC MSCs reduced the level of IL-6 by 80.0% 2 days after treatment and by 93.4% at the endpoint	Relief of RA disease symptoms in a CIA model	[136]
CIA in DBA/1 mice	hAT MSCs	IV injection on day 28 after arthritis induction for the next five days	2 × 10^6^ cells	hAT MSCs induced the expansion of Tregs both in the peripheral blood and spleen (in vivo); and downregulated the level of TNF-α, IL-1β and IL-6 in mouse macrophages and inhibited the proliferation of human primary T cells (in vitro)	Attenuation of systemic inflammation in mice with CIA	[119]
CIA in Balb/c mice	Murine BM MSCs	IV injection of MSCs and IP injection of IL-4 at day 21	5 × 10^6^ cells	BM MSCs in combination with IL-4 treatment decreased the levels of RF, C-reactive protein (CRP) and anti-nuclear antibodies; TNF-α and monocyte chemoattractant protein-1 (MCP-1) levels. Additionally, BM MSCs decreased the levels of cartilage oligomeric matrix protein (Comp), tissue inhibitor metalloproteinase-1 (Timp1), MMP-1 and IL-1 receptor	Reduction of joint inflammation, synovial cellularity, vascularization and bone destruction in a CIA model	[144]
CIA in female Wistar rats	hUC MSCs	IP injection on days 16 and 18	2 × 10^6^ cells	hUC MSCs downregulated the functions of activated CD4+ and CD8+ T cells, suppressed the secretion of pro-inflammatory cytokines and induced the expansion of Tregs	Slowing down the progression of disease activity	[140]

**Table 3 ijms-22-11592-t003:** Clinical trials for RA treatment with MSCs.

Clinical Trial Identifier	Study Design	Cell Source	Number of Patients	Route of Administration and Doses	Follow-Up Time (Months)	Clinical Status before Treatment or Control Group	Clinical Status after Treatment	Reference
NCT01663116	Randomized, multicenter, double-blind, placebo-controlled, dose-escalation phase Ib/IIa	Allogeneic AT MSCs	53	1, 2 or 4 × 10^6^ cells/kg of body weight, three IV injections, weekly	6	DAS28-ESR↑, CRP↑, ACR20 response after 1 month (29%) and 3 month (0%)	DAS28-ESR↓, CRP↓, ACR20 response after 1 month (20–45%) and 3 month (15–25%)	[16]
Unknown	Pilot	Autologous AT MSCs	3	Patient 1: two separate IV injections of 3 × 10^8^ cells, 15 week interval Patient 2: once 2 × 10^8^ cells (IV injection) + 1 × 10^8^ cells (IA injection); once 3.5 × 10^8^ cells (IV injection) + 1.5 × 10^8^ cells (IA injection), 3-month interval Patient 3: four separate IV injection of 2 × 10^8^ cells, 4-week interval	3–13	VAS↑, KWOMAC↑, CRP↑, RF↑, anti-CCP↑, Standing time↓, WD↓	VAS↓, KWOMAC↓, CRP↓, RF↓, anti-CCP↓, standing time↑, WD↑, off steroids	[17]
NCT03333681	Phase I	Autologous BM MSCs	9	1 to 2 × 10^6^ cells/kg of body weight, single IV injection	12	DAS28-ESR↑, VAS↑, ESR↑, CRP↑, RF↑, anti-CCP↑	DAS28-ESR↓, VAS↓, ESR↓, CRP↓(NS), RF↓, anti-CCP↓ (NS)	[18]
NCT01873625	Randomized, triple-blind, single-center, placebo-controlled phase I/II	Autologous BM MSCs	30	4.2 × 10^7^ cells/patient, single IA injection	12	DAS28↑, VAS↑, WOMAC↑, ESR↑, CRP↑, Pain FWD↓, WD↓, Time to jelling↓, Standing time↓	DAS28↓ (NS), VAS↓, WOMAC↓, ESR↓ (NS), CRP↓ (NS), Pain FWD↑, WD↑, Time to jelling↑, Standing time↑	[19]
NCT01547091	Prospective phase I/II	Allogeneic UC MSCs	172	4 × 10^7^ cells/patient, single IV injection	36	DAS28↑, HAQ↑, CRP↑, ESR↑, RF↑, anti-CCP↑, TNF-α↑, IL-6↑	DAS28↓, HAQ↓, CRP↓, ESR↓, RF↓, anti-CCP↑, TNF-α↓, IL-6↓	[20,21]
NCT02221258	Phase Ia, open-label, dose-escalation	Allogeneic UCB MSCs	9	2.5 × 10^7^, 5 × 10^7^, or 1 × 10^8^ cells/patient, single IV injection	1	DAS28↑, VAS↑, HAQ↑, CRP↑, IL-1β↑, IL-6↑, IL-8↑, TNF-α↑	DAS28↓, VAS↓, HAQ↓, CRP↓, IL-1β↓, IL-6↓, IL-8↓, TNF-α↓	[22]

Western Ontario and McMaster Universities Arthritis Index (WOMAC); Korean Western Ontario and McMaster Universities Arthritis Index (KWOMAC); visual analogue scale (VAS); the American College of Rheumatology criteria (ACR); Health Assessment Questionnaire (HAQ); disease activity score 28 (DAS28); pain-free walking distance (Pain FWD); walking distance (WD); erythrocyte sedimentation rate (ESR); C-reactive protein (CRP); rheumatoid factor (RF); anti-cyclic citrullinated antibody (anti-CCP); non-significant (NS); increasing level (↑); decreasing level (↓).

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
