# Peer review of "Mesenchymal Stem Cell-Based Therapy for Rheumatoid Arthritis"

_ijms, 2021, doi:10.3390/ijms222111592_

Round 1

Reviewer 1 Report

This is a resubmission with essentially the same scope as the previously submitted manuscript.  The authors have adequately covered the area of the topic however the topic is quite broad and new sections have essentially expanded some previously briefly described points.  

The section on the current treatments for RA is unnecessarily long - almost 2 pages (plus a 2 page table) - when 2 paragraphs would have sufficed.

Further examination of the role of apoptosis in the therapeutic mechanism of MSC would be desirable.  Additionally, the role of IDO1 is mentioned in passing but has more than a few published studies implicating its role (e.g. PMID: 24452999, PMID: 27008974).

Author Response

Dear Reviewer, 

Thank you for your thorough review.

Please find our reply in the attachment.

Thank you.

Best regards,

Authors 

Reviewer 2 Report

The authors have addressed my comments, and publication of the manuscript is recommended.

Author Response

Dear Reviewer,

Thank you for your thorough review and the recommendation for publication. 

Please find our reply in the attachment. 

Best regards,

Authors

This manuscript is a resubmission of an earlier submission. The following is a list of the peer review reports and author responses from that submission.

Round 1

Reviewer 1 Report

The authors have collected a comprehensive set of articles describing the therapeutic use of mesenchymal cells in the treatment of RA. In general, the data is referenced correctly, but not interpreted to bring added value. Instead of repeating the slogans from published work, it would be important to critically evaluate the clinical and biological impact of each presented mechanism and study outcome.

  1. Intro: The age of onset for RA is an underestimation, it is usually initially diagnosed in older patients mean age being above 50 years. Considering the improved therapeutic options, the disability figures sound quite high. Please find references to support all claims in the first part of the introduction. Reference number 1 is no longer available online, please check.

Seronegative and positive RA are different, not only potentially different. Although the different disease forms are treated in the same way, there are significant differences in the disease biology between the two disease forms. Please, check the literature and revise accordingly.

  1. Treatment: The treatments should be discussed in order of significance. First methotrexate, then corticosteroids and biologicals. NSAIDs are used, but not so much to control the destructive inflammation.

The authors have focused on the biochemical effects of the drugs. However, as RA is an immune-mediated disease, also the immunological consequences and mechanisms should be systematically addressed to get the big picture correct.

I feel, the authors are not very familiar with RA and associated biology and, thus, I suggest the authors discuss this part of the introduction with an experienced rheumatologist.

3. MSC in RA treatment:

3.1 Preclinical studies:

-This part should be divided so that it is clear which parts originate from mouse models and what parts describe findings from analyses performed using human sample material. It is important to never refer to CIA data as RA data. Both are important but should be interpreted in their correct contexts; clearly separating mouse from man.

3.2. Clinical studies:

-Suggestion: summary table to allow quick evaluation of the studies containing the core information (study design, serostatus of the included patients, to which treatment they are refractory, safety, impact on disease, follow-up time, administration route and how many injections, number of patients, control group information…)

- The studies are technically well summarized, but the results are not interpreted. Based on the presented data, MSC therapy is well-tolerated, which is not surprising as it is performed using autologous cells. The clinical efficacy, however, should be critically discussed. It is important to discuss whether the observed improvements in the clinical status are clinically or only statistically significant. Discuss the different readouts reported in different studies. The most commonly used metrics include DAS28, CRP, ESR, Number of affected joints, ACR20, ACR50… Many of the reported studies used very unconventional parameters. The reasons behind and the limitations of these innovative measures should be discussed. Also, whether the observed improvement in clinical recordings is in balance with the cost of the very laborious and expensive treatment procedure should be discussed.

So, please do not only summarize what has been observed in different studies but add a layer of critical interpretation to make clinically relevant and holistic conclusions.

3.3 Improvement strategies:

-Suggestion: An illustration to summarize the different proposed strategies?

Other questions:
In which diseases have the MSCs been therapeutically most efficient? Which of these are inflammatory? Autoimmune? Local or systemic? Based on these responses one could deduce what kind of disease mechanisms are best affected by these cells.

Author Response

Dear Reviewer,

Best regards, 

Authors 

Reviewer 2 Report

The review manuscript entitled “Mesenchymal Stem Cell-based Therapy for Rheumatoid Arthritis Treatment” by Sarsenova and colleagues provides a detailed description of the use of Mesenchymal stem cells (MSCs) as therapeutic tools for the treatment of Rheumatoid arthritis. The first section ofd the manuscript provides detailed information on the current approaches in Rheumatoid arthritis treatment and the role of mesenchymal Stem Cells for Rheumatoid arthritis terapy. The remaining sections of the ms describe the most important preclinical and clinical studies based on Mesenchymal stem cells-therapy, while additional Mesenchymal stem cells-based Rheumatoid arthritis therapy strategies are also provided. The manuscript is well organized and well written in general. It will be interesting and attractive for the reader. In my opinion, it should be accepted after a minor revision. I have several observations for improving the manuscript 

Main points
1.    The manuscript is interesting and attractive for the reader. However, the section describing the most important clinical trials can be improved by including a table summarizing the most important clinical studies based on Mesenchymal Stem Cell-based therapies. It will be helpful for the reader. In addition, the IDs (FROM clinicaltrials.gov) of the clinical trials described in the text (refs 95-101) should be included 
2.    The most important references in the field has been included. However, several sentences in the manuscript are lacking in supporting references (especially in the introductive paragraphs), I have suggested some references (see below).
3.    Figure 1 should be uploaded with a higher quality

Minor comments
Lines 37-39 This sentence is lacking in supporting references. A detailed information regarding the risk factors linked to Rheumatoid arthritis can be found here PMID: 29736302
Lines 55 and lines 118-120 Alongside the development of the already mentioned liquid tumors, an additional side effect related to use of DMARDs for the treatment of Rheumatoid arthritis is the onset of skin tumors such as Merkel cell carcinoma PMID: 28174236
Line 77 NSAIDs and GCs, should be reported as Non-steroidal anti-inflammatory drugs and glucocorticoids, respectively, when quoted for the first itme.
lines 89-92  This sentence is lacking in supporting references. For instance PMID: 19095768
Line 161 Mesenchymal Stem Cells have also been isolated from the umbilical cord (other than the already quoted umbilical cord blood), as reported here PMID: 33898434. For completeness of information, this finding/ref should be included 
Line 222 please detail the meaning of DCs
Lines 407-410 please include references supporting the mentioned strategies designed to improve the immunomodulatory and anti-inflammatory properties of mesenchymal stem cells in RA. For instance PMID: 32784608
Lines 434-430 These sentences are lacking in supporting references. For instance PMID: 27847522

Author Response

Dear Reviewer,

Best regards, 

Authors 

Reviewer 3 Report

I have had the pleasure of reviewing the review of Sarsenova M et. al. which summarises MSC based therapy for RA.  Overall, the manuscript sufficiently covers the emerging field of MSC-based treatment of chronic disease.
Below are some suggestions.

1. The authors may wish to consider dropping the word 'Treatment' from their manuscript title as it is made somewhat redundant by the preceeding use of the word 'Therapy'.

2. Ln 30 comma goes after the 'and' not before.

3. Ln 105 "Methotrexate (MTX)...second-line drug."  Consider rephrasing or removing the second 'and'.

4. Ln 114-115 (and Ln 125-138) Consider adding TM or R symbol to drug names.

5. Ln 179 Consider adding a semicolon after inflammation. E.g. "...sites of inflammation; these chemokines bind to..."

6. Ln 193-196 "Macrophages play...RA patients."  Consider citing - Mulherin D, Fitzgerald O, Bresnihan B. Synovial tissue macrophage populations and articular damage in rheumatoid arthritis. Arthritis Rheum. 1996;39:115-124.

7. Add a relevant reference to the end of the sentence ending at Ln 271.

8. Ln 272 "...have a potential in reducing..." Consider rephrasing.

9. Ln 412.  The comma after "Genes" may not convey the intended meaning of the sentence.

10. Please proof-read carefully.

11. The authors may wish to supplement their discussion with the following studies, particularly Galleu et al.

Nicolaidou et al. Monocytes induce STAT3 activation in human mesenchymal stem cells to promote osteoblast formation. 
PLoS One. 2012;7(7):e39871. doi: 10.1371/journal.pone.0039871.  

Galleu A et al. Apoptosis in mesenchymal stromal cells induces in vivo recipient-mediated immunomodulation Sci Transl Med. 2017 Nov 15;9(416):eaam7828. 

Alavi M et al. Intravenous injection of autologous bone marrow-derived mesenchymal stem cells on the gene expression and plasma level of CCL5 in refractory rheumatoid arthritis. J Res Med Sci. 2020 Dec 30;25:111. 

Author Response

Dear Reviewer,

Best regards, 

Authors 

Round 2

Reviewer 1 Report

Please find my comments in the attached document. 

Author Response

Dear Editors,

Please find the response to Reviewer attached. 

Best regards,

Authors

Round 3

Reviewer 1 Report

I still encourage the authors to include a rheumatologist in the manuscript preparation team. That would significantly improve the quality of the ms.

Intro

In reference 1, the authors display a rough schematic presentation of the timeline of RA. This is not even meant to be an accurate presentation to conclude the age of onset of RA. Thus, the claim that ”pain and deformity between the ages of 50 and 60” cannot be based on that reference. Especially when the modern treatments are taken into account. The claim would stand if it specifically referred to untreated disease. Please, report a realistic figure about the age of onset. Pain is observed decades before actual deformations develop.

The second paragraph starting from r 36. Text is descriptive and stays on a very general level.

Seronegative RA is an array of different conditions characterized by joint inflammation including many misdiagnosed patients with eg. polymyalgia rheumatica, psoriatic arthritis, etc. Also, some seroconvert after the onset of clinical symptoms. (PMID: 29998832) And, additionally, there is no clear evidence that the real seroneg RA would be a less severe condition than the seropos.

The treatment part of the Introduction (starting from row 56) section is still unclear and the treatment parts, in general, would benefit from some kind of illustration or table. Although much improved, the RA introduction still needs further attention.

Although much better, also the MSC sections tend to be lists of specific details. Details themselves are not bad, but they should be woven into a story.

Also, the use of the English language should be improved. A collaboration with a professional writer, who knows how to express things clearly and precisely would probably improve the reader experience. In its present form, I am afraid many readers stop reading as the general presentation does not meet the rigorous standards of modern scientific literature.